# Simplify to Amplify: Achieving Information-Theoretic Bounds with Fewer Steps in Spectral Community Detection

## Abstract

We propose a streamlined spectral algorithm for community detection in the two-community stochastic block model (SBM) under constant edge density assumptions. By reducing algorithmic complexity through the elimination of non-essential preprocessing steps, our method directly leverages the spectral properties of the adjacency matrix. We demonstrate that our algorithm exploits specific characteristics of the second eigenvalue to achieve improved error bounds that approach information-theoretic limits, representing a significant improvement over existing methods. Theoretical analysis establishes that our error rates are tighter than previously reported bounds in the literature. Comprehensive experimental validation confirms our theoretical findings and demonstrates the practical effectiveness of the simplified approach. Our results suggest that algorithmic simplification, rather than increasing complexity, can lead to both computational efficiency and enhanced performance in spectral community detection.

## 1 Introduction

Community detection represents a fundamental challenge in statistics, theoretical computer science, and image processing. The stochastic block model (SBM) serves as a prominent theoretical framework for analyzing this problem. In its simplest form, the model consists of two equal-sized blocks $V_1$ and $V_2$, each containing $n$ vertices. A random graph is generated according to the following distribution: edges between vertices within the same block occur with probability $\frac{a}{n}$, while edges between vertices in different blocks occur with probability $\frac{b}{n}$, where $a > b > 0$. Given such a graph, various algorithms exist for block recovery Chin et al. (2015), Bui et al. (1984), Dyer & Frieze (1989), McSherry (2001), Coja-Oghlan (2009).

In the sparse graph case, with high probability, the graph contains a linear fraction of isolated vertices Bollobás (2001). Since these isolated vertices lack connectivity information, perfect recovery of the community structure is impossible. However, we can still accurately recover a substantial portion of each block. Formally, we would like to find a partition of $V_1', V_2'$ of $V = V_1 \cup V_2$ such that $V_i$ and $V_i'$ are very close to each other. To quantify the recovery accuracy, we introduce the following definition:

**Definition 1.1.** *A collection of subsets $V_1', V_2'$ of $V_1 \cup V_2$ is $\gamma$-correct if $|V_i \cap V_i'| \geq (1-\gamma)n, i = 1, 2$.*

We would like to devise an algorithm that can guarantee $\gamma$-correctness for small $\gamma$ with high probability in polynomial time. In Coja-Oghlan (2009), Coja-Oglan proved

**Theorem 1.2.** *For any constant $\gamma > 0$, there exist constants $C_1, C_2 > 0$ such that if $a, b > C_1$ and $\frac{(a-b)^2}{a+b} > C_2 \log(a+b)$, one can find a $\gamma$-correct partition using a polynomial time algorithm.*

In Chin et al. (2015), Chin et al. introduced a Spectral Algorithm that achieves exponential bounds on the incorrect recovery rate in the case of a sparse graph.

**Theorem 1.3.** *There are constants $C_1, C_2 > 0$ such that the following holds. For any constants $a > b > C_1$ and $\gamma > 0$ satisfying*

$$\frac{(a-b)^2}{a+b} \geq C_2 \log \frac{2}{\gamma} \tag{1}$$

*one can find a $\gamma$-correct partition with probability $1 - o(1)$ using a simple spectral algorithm.*

Theorem 1.3 improves the relation between the accuracy $\gamma$ and the ratio $\frac{(a-b)^2}{a+b}$. Moreover, this bound is asymptotically sharp because according to Zhang & Zhou (2015), there exists a constant $c > 0$ such that when

$$\frac{(a-b)^2}{a+b} \leq c \log \frac{1}{\gamma} \tag{2}$$

one **cannot** recover a $\gamma$-correct partition (in expectation), regardless of the algorithm.

The standard Spectral Algorithm comprises two stages: **Spectral Partition** and **Correction** (detailed in Section 2). Previous work established that Spectral Partition alone achieves only inverse-square correctness rates, requiring the Correction step to reach the desired inverse-log relationship. However, our experiments reveal that Spectral Partition actually produces inverse-log performance without correction, suggesting this additional step is unnecessary.

Our theoretical analysis identifies a non-tight lemma in the original proof that underestimates the algorithm's performance. We provide improved bounds and experimentally demonstrate that these bounds are sharp, eliminating the need for the Correction step to achieve the inverse-log rates claimed in Chin et al. (2015). Additionally, we streamline the Spectral Partition itself by removing redundant operations, ensuring that the resulting vectors maintain statistical independence, a property that will prove valuable for future algorithmic improvements (discussed in Section 5).

The rest of this paper is organized as follows: Section 2 presents the original Spectral Algorithm and our simplified version. Section 3 shows that our simplification maintains and improves theoretical bounds. Section 4 validates our predictions experimentally. Section 5 summarizes our findings and discusses future work.

## 2 ORIGINAL SPECTRAL ALGORITHM

In Chin et al. (2015), Chin et al. gave the Spectral Algorithm that guarantees the result in Theorem 1.3. But first let us define some variables. Let $A$ denote the adjacency matrix of a random graph generated from the distribution described in Section 1. And let $A_E = \mathbb{E}[A]$ be the expected adjacency matrix, with entries $a/n$ and $b/n$. Then $A_E$ is a rank two matrix with two non-zero eigenvalues $\lambda_1 = a + b$ and $\lambda_2 = a - b$. Then unit eigenvector $\boldsymbol{u_1}$ corresponding to the eigenvalue $a + b$ has coordinates:

$$\boldsymbol{u_1}(i) = \frac{1}{\sqrt{2n}} \forall i = 1, \dots, 2n \tag{3}$$

while the unit eigenvector $\boldsymbol{u_2}$ corresponding to the eigenvalue $a - b$ has coordinates

$$\boldsymbol{u_2}(i) = \begin{cases} \frac{1}{\sqrt{2n}} & \text{if } i \in V_1 \\ -\frac{1}{\sqrt{2n}} & \text{if } i \in V_2 \end{cases} \tag{4}$$

---

**Spectral Partition.**

1. Input the adjacency matrix $A, d := a + b$.

2. Zero out all the rows and columns of $A$ corresponding to vertices whose degree is bigger than $20d$, to obtain the matrix $A'$.

3. Find the eigenspace $W$ corresponding to the top two eigenvalues of $A'$.

4. Compute $\boldsymbol{v_1}$, the projection of all-ones vector on to $W$

5. Let $\boldsymbol{v_2}$ be the unit vector in $W$ perpendicular to $\boldsymbol{v_1}$.

6. Sort the vertices according to their values in $\boldsymbol{v_2}$, and let $V_1' \subset V$ be the top $n$ vertices, and $V_2' \subset V$ be the remaining $n$ vertices

7. Output $(V_1', V_2')$.

---

Figure 1: Spectral Partition

The second eigenvector $u_2$ of the expected adjacency matrix $A_E$ encodes the true community structure. Let $w_1$ and $w_2$ denote the first and second eigenvectors of the observed adjacency matrix $A$, respectively. Our goal is to use $w_2$ as a proxy for the unknown $u_2$. The **Spectral Algorithm** in Figure 1 produces vector $v_2$ that closely approximates $u_2$, achieving the following result:

**Theorem 2.1.** *There are constants $C_1, C_2 > 0$ such that the following holds. For any constants $a > b > C_1$ and $\gamma > 0$ satisfying*

$$\frac{(a-b)^2}{a+b} \geq C_2 \frac{1}{\gamma^2} \tag{5}$$

*one can find a $\gamma$-correct partition with probability $1 - o(1)$ using* **Spectral Partition.**

The bound in Theorem 2.1 is weaker than that claimed in Theorem 1.3. To achieve the inverse-log relationship, the original work requires a second **Correction** step (Figure 2), yielding the complete algorithm shown in Figure 3. The correction mechanism works as follows: provided **Spectral Partition** achieves sufficiently low error rate $\gamma$, the **Correction** step reduces this to exponentially small values.

---

**Correction.**

1. Input: a partition $V_1', V_2'$ and a Blue graph on $V_1' \cup V_2'$.

2. For any $u \in V_1'$, label $u$ *bad* if the number of neighbors of $u$ in $V_2'$ is at least $\frac{a+b}{4}$ and *good* otherwise.

3. Do the same for any $v \in V_2'$.

4. Correct $V_i'$ be deleting its bad vertices and adding the bad vertices from $V_{3-i}'$.

---

Figure 2: Correction

Specifically, Lemma 2.3 in Chin et al. (2015) establishes that if the input to **Correction** is $c$-correct for some $c > 0$, then the output achieves $\gamma$-correctness with $\gamma = 2\exp\left(-f(c)\frac{(a-b)^2}{a+b}\right)$ where $f(c) > 0$ depends only on $c$. The complete two-stage algorithm of Chin et al. is therefore the **Partition** procedure in Figure 3.

---

**Partition**

1. Input the adjacency matrix $A$, $d := a + b$.

2. Randomly color the edges with Red and Blue with equal probability.

3. Run **Spectral Partition** on Red graph, outputting $V_1', V_2'$.

4. Run **Correction** on the Blue graph.

5. Output the corrected sets $V_1', V_2'$.

---

Figure 3: Partition

## 2.1 OUR MODIFIED ALGORITHM

Our key modification to **Spectral Partition** eliminates step 2, which zeros out rows and columns corresponding to vertices with degree greater than $20d$. Instead, we work directly with the original adjacency matrix $A$ throughout the algorithm. While this preprocessing step was essential for two lemmas in the original analysis, it destroys the statistical independence of matrix entries in $A'$. By working with $A$ directly, we preserve the independent distribution of matrix entries and can subsequently maintain independence in the entries of eigenvector $w_2$. This independence property proves crucial for our analysis in Section 3 and may help future algorithmic enhancements we explore in Section 5.

The first lemma requiring step 2 is restated in Theorem 2.2. Define $M = A - A_E$ as the difference between the observed and expected adjacency matrices. Let $M'$ denote the matrix obtained by applying the same row and column deletions to $M$ as performed on $A$ in step 2 of **Spectral Partition**. Chin et al. (2015) establish the following result:

**Theorem 2.2.** *There exist constants $C_1, C_2$ such that if $a > b > C_1$, and matrix $M'$ is obtained as described above, then we have*

$$||M'|| \leq C_2 \sqrt{a + b} \qquad (6)$$

*with probability $1 - o(1)$.*

Throughout this paper, $||M'||$ denotes the spectral norm $\sup\{||Mx||_2 : ||x||_2 \leq 1\}$, and all matrix norms follow this convention. While the original proof of Theorem 2.2 depends on the deletion step, we show that the bound holds without deletion, with only modest increases in the constants $C_1, C_2$. Our proof, which leverages techniques from Füredi & Komlos (1981) and Krivelevich & Vu (2000), is provided in the appendix.

The second lemma that depends on the deletion step appears in the **Correction** step analysis. Since our simplified algorithm eliminates this step entirely, we don't have to analyze the implications of our modification to this step.

## 3 IMPROVED ERROR BOUNDS FOR SPECTRAL PARTITION

### 3.1 ORIGINAL ERROR BOUNDS

Let $W$ be the two-dimensional eigenspace corresponding to the top two eigenvalues of $A$, and let $W_E$ be the corresponding eigenspace of $A_E$. Chin et al. Chin et al. (2015) establish that the angle $\angle(W, W_E)$ between these subspaces is sufficiently small, where we use the standard convention $\sin \angle(W_1, W_2) := ||P_{W_1} - P_{W_2}||$ with $P_W$ denoting the orthogonal projection onto subspace $W$.

As a consequence of this subspace proximity, the angle between $\boldsymbol{u_2}$ (the second eigenvector of $A_E$) and $\boldsymbol{v_2}$ (the vector obtained in step 5 of **Spectral Partition**) is also small. The key insight is that when these vectors are well-aligned, **Spectral Partition** produces an accurate community assignment. Specifically, the analysis in Chin et al. (2015) bounds $\sin \angle(\boldsymbol{u_2}, \boldsymbol{v_2})$ and establishes the following result:

**Theorem 3.1.** *There exist constants $C_1, C_2$ such that if $a > b > C_1$, and vectors $\boldsymbol{u_2}, \boldsymbol{v_2}$ are as described above, then we have*

$$\sin \angle(\boldsymbol{u_2}, \boldsymbol{v_2}) \leq C_2 \sqrt{\frac{\sqrt{a+b}}{a-b}} \qquad (7)$$

*with probability $1 - o(1)$.*

Finally, Chin et al. (2015) shows that $\gamma \leq \frac{4}{3} \sin^2 \angle(\boldsymbol{u_2}, \boldsymbol{v_2})$, which gives us the following result:

**Theorem 3.2.** *There exist constants $C_1, C_2$ such that if $a > b > C_1$, then we have*

$$\gamma \leq C_2 \frac{\sqrt{a+b}}{a-b} \qquad (8)$$

*with probability $1 - o(1)$.*

which proves Theorem 2.1.

Our experiments reveal that Theorem 3.1 is tight, while Theorem 3.2 is not. In general, Theorem 3.2 is indeed sharp. There exist vectors $\boldsymbol{u_2}, \boldsymbol{v_2}$ achieving equality up to a constant factor. However, the **Spectral Algorithm** produces vectors $\boldsymbol{v_2}$ with specific structural properties that render this bound loose. We prove that under these properties, significantly tighter bounds are achievable.

### 3.2 SHARPNESS OF THEOREM 3.2

To establish the sharpness of Theorem 3.2, we formulate the following optimization problem. Let $x_1, \ldots, x_{2n}$ denote the entries of $\boldsymbol{v_2}$ with $\sum x_i^2 = 1$ and $x_1 \geq \cdots \geq x_{2n}$. The partition step assigns

indices $\{1, \ldots, n\}$ to community $V_1$ and $\{n+1, \ldots, 2n\}$ to community $V_2$. For fixed error rate $\gamma$, let $k = \gamma n$ (assuming $k$ is integer), representing the number of misclassified vertices.

Our goal is to minimize the angle $\theta = \angle(\boldsymbol{u_2}, \boldsymbol{v_2})$ subject to fixed $\gamma$, equivalent to maximizing $\cos\theta$. The true community indicator satisfies $w_i = 1/\sqrt{2n}$ for $i \in V_1$ and $w_i = -1/\sqrt{2n}$ for $i \in V_2$. Without misclassification:

$$\cos\theta = \sum_{i=1}^{2n} x_i w_i = \frac{1}{\sqrt{2n}} \left( \sum_{i=1}^{n} x_i - \sum_{i=n+1}^{2n} x_i \right)$$

To maximize $\cos\theta$ under exactly $k$ misclassifications, the optimal strategy places errors among entries with smallest magnitudes. Specifically, vertices $\{n-k+1, \ldots, n\}$ from $V_1$ are misassigned to $V_2$, while vertices $\{n+1, \ldots, n+k\}$ from $V_2$ are misassigned to $V_1$, yielding:

$$\cos\theta \leq \frac{1}{\sqrt{2n}} \left( \sum_{i=1}^{n-k} x_i - \sum_{i=n-k+1}^{n} x_i + \sum_{i=n+1}^{n+k} x_i - \sum_{i=n+k+1}^{2n} x_i \right) \tag{9}$$

This bound is achieved by the assignment $x_1 = \cdots = x_{n-k} = 1/\sqrt{2(n-k)}$, $x_{n-k+1} = \cdots = x_{n+k} = 0$, and $x_{n+k+1} = \cdots = x_{2n} = -1/\sqrt{2(n-k)}$, which satisfies the normalization constraint and yields $\cos\theta = \sqrt{1-\gamma}$. Therefore $\gamma = \sin^2\theta$, confirming that Theorem 3.2 is sharp up to constants.

### 3.3 STATISTICAL PROPERTIES OF THE SECOND EIGENVECTOR

Abbe et al. (2019) demonstrate that the second eigenvector can be approximated as $\boldsymbol{w_2} \approx \frac{A\boldsymbol{u_2}}{a-b}$ with error bound $||\boldsymbol{w_2} - \frac{A\boldsymbol{u_2}}{a-b}||_\infty = o(1/\sqrt{n})$

The denominator $a - b$ is irrelevant as $\boldsymbol{w_2}$ will be scaled to be a unit vector. Thus we now focus on characterizing the distribution of $A\boldsymbol{u_2}$. For vertex $i \in V_1$, the $i$-th entry of $A\boldsymbol{u_2}$ equals the difference between the number of edges between $i$ and vertices in $V_1$, and the number of edges between $i$ and vertices in $V_2$. Since each edge appears independently with probability $a/n$ (within-community) or $b/n$ (between-community), this entry follows the distribution of a difference of two binomial random variables. Specifically, let

$$Y \sim \text{Binomial}(n, a/n) - \text{Binomial}(n, b/n) \tag{10}$$

Then each entry of $A\boldsymbol{u_2}$ is distributed as $Y$ or $-Y$ with equal probability, depending on whether $i \in V_1$ or $i \in V_2$.

### 3.4 APPLYING CHERNOFF BOUNDS TO RELATE $\gamma$ AND $\sin\theta$

Building on the optimization framework above, while we know the approximate distribution of the $x_i$ entries, direct analysis remains computationally intractable. Instead, we leverage constraints derived from Chernoff concentration inequalities applied to this distribution. The Chernoff bound states that for a random variable $X$ with moment generating function $M(t)$:

$$P(X \geq a) \leq M(t)e^{-ta} \quad \forall a, \forall t > 0$$

This bound becomes increasingly sharp in the tail regions for large values of $a$. For approximately bell-shaped distributions, Chernoff bounds at multiple points constrain the distribution's tail behavior, effectively providing lower bounds on how "concentrated" the distribution must be around its center.

Applied to our ordered sequence $x_1 \geq x_2 \geq \cdots \geq x_n$, these concentration properties impose lower bounds on the decay rates between consecutive entries:

$$\frac{x_1}{x_2}, \frac{x_2}{x_3}, \ldots, \frac{x_{n-1}}{x_n}$$

Define $p_a = a/n$, $q_a = 1 - p_a$, $p_b = b/n$, $q_b = 1 - p_b$, and the optimal Chernoff parameter $t^* = \frac{1}{2}\ln\left(\frac{p_a q_b}{q_a p_b}\right)$. Let the concentration constant be:

$$C = \frac{1}{2}(\sqrt{p_a p_b} + \sqrt{q_a q_b})^{2n} + \frac{1}{2}\left(\sqrt{\frac{q_a^3 p_b^3}{p_a q_b}} + \sqrt{\frac{p_a^3 q_b^3}{q_a p_b}} + q_a q_b + p_a p_b\right)^n$$

The Chernoff concentration inequalities translate into the following optimization constraints:

$$x_1^2 + \cdots + x_{2n}^2 \leq 1$$

$$x_{i+1} \leq \frac{\ln C + \ln(2n+1) - \ln(i+1)}{\ln C + \ln(2n+1) - \ln i} x_i \quad \forall i = 1, \ldots, n-1$$

$$x_i \geq \frac{\ln C + \ln(2n+1) - \ln(2n+1-i)}{\ln C + \ln(2n+1) - \ln(2n-i)} x_{i+1} \quad \forall i = n+1, \ldots, 2n-1$$

The complete derivation appears in the appendix. Since $C$ is known before any optimization, these constraints together with Equation 9 as the objective function define a convex optimization problem. We solve this optimization problem umerically to find the maximum value of $\cos\theta$ subject to the above constraints. Our theoretical analysis predicts this maximum should satisfy (proof in the appendix):

$$\cos\theta \leq \frac{\sqrt{2n}}{t^*}(1-\gamma)\left(\ln C + 1 + \ln\frac{2 + \frac{1}{n}}{1-\gamma}\right) \tag{11}$$

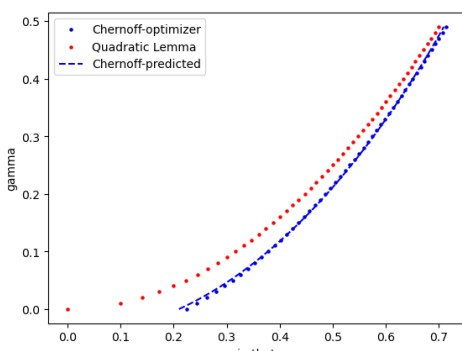 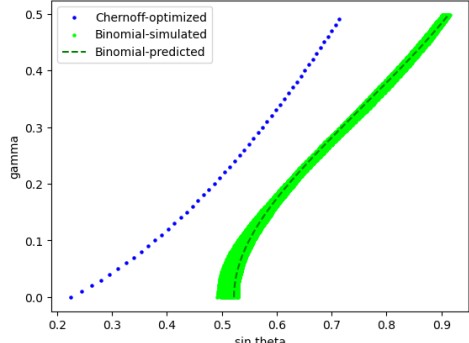

(a) $\gamma$ as a function of $\sin\theta$: Theorem 3.2 and Chernoff-derived bounds

(b) $\gamma$ as a function of $\sin\theta$: Chernoff-derived bounds and Monte Carlo / Normal approximations

Figure 4: $\gamma$ as a function of $\sin\theta$ for various approaches

Figure 4a presents our experimental validation results for $n = 500$, $a = 0.06n$, $b = 0.04n$. The red points represent the relationship from Theorem 3.2, while the blue points show the actual optimization results under our Chernoff-derived constraints. The blue line displays our theoretical prediction from Equation 11, fitted to the optimization data using ordinary least squares (OLS) regression to account for the unit normalization of the $x_i$ vector.

The results demonstrate that our Chernoff-based analysis yields significantly tighter bounds than the original theorem. For any given value of $\sin\theta$, our approach provides a substantially lower upper bound on the achievable error rate $\gamma$. Furthermore, the close agreement between the blue line and blue points confirms the accuracy of our theoretical prediction in Equation 11.

### 3.5 Monte-Carlo Simulation and Normal Approximation to Relate $\gamma$ and $\sin\theta$

Given the distribution in Equation 10, we can directly generate samples of the $x_i$ entries using Monte Carlo methods, removing the need for numerical optimization. With the $x_i$ values generated from their distribution, Equation 9 provides the maximum $\cos\theta$ for any given error level $k$.

While we could compute Equation 9 directly using the exact probability density function, this approach is algebraically intractable. Instead, we use a normal approximation to simplify the analysis. The binomial distributions in our model satisfy the standard approximation conditions: both $np \geq 20$ and $n(1-p) \geq 20$ hold for our parameter ranges, so that the approximation is reasonable.

Under this normal approximation, the difference of binomials $Y$ also approaches normality, and consequently each entry $X_i$ becomes approximately normal. This normality assumption enables us to derive a closed-form theoretical prediction for the performance bound. Using the normal approximation and the structure of our optimization problem, we obtain the following theoretical prediction (with derivation provided in the appendix):

$$\cos\theta \leq \frac{2}{\sqrt{2n}}(2n+1)\left(2\phi\left(-\Phi^{-1}\left(\frac{1-\gamma}{2+1/n}\right)\right) - \phi\left(-\Phi^{-1}\left(\frac{1}{2+1/n}\right)\right)\right) \tag{12}$$

where $\phi$ and $\Phi$ denote the standard normal probability density function and cumulative distribution function, respectively. In the derivation above, we assumed that the entries $x_i$ follow a standard normal distribution with mean 0 and unit variance. While the zero-mean assumption is valid, the unit variance assumption is not. The actual entries will have a different variance determined by the underlying binomial distributions and the problem parameters. However, since the final vector must satisfy the normalization constraint $\sum x_i^2 = 1$, the entries will be appropriately scaled regardless of their original variance. The theoretical prediction in Equation 12 captures the correct functional relationship between $\gamma$ and $\cos\theta$, but with a scaling factor that depends on the actual variance of the entries.

Figure 4b presents our experimental validation using the same parameters as before: $n = 500$, $a = 0.06n$, $b = 0.04n$. We conducted Monte Carlo simulations with 50 repetitions to minimize random variation in our results. The green points represent the $(\sin\theta, \gamma)$ pairs computed from each simulation run, forming a "band" due to the natural clustering of results across repetitions. The green dashed line shows our theoretical prediction from Equation 12, fitted to the simulation data using OLS regression to account for the normalization constraint. For comparison, we include the blue points from our earlier Chernoff-based analysis (Section 3.4, Figure 4a). The results validate several important aspects of our theoretical framework:

First, the close agreement between the green dashed line and the simulation points confirms that our normal approximation in Equation 12 accurately captures the underlying relationship between error rate and spectral alignment.

Next, the green band lies well below the blue points, demonstrating that while our Chernoff-derived bounds are mathematically sound, they remain conservative estimates. The gap between these approaches becomes particularly pronounced for small error rates, precisely the region most relevant for practical applications. This suggests that the Chernoff bounds, though tight in a worst-case sense, do not fully capture the distributional properties that emerge in typical use cases.

Perhaps most significantly, both our simulation and Chernoff analysis reveal that perfect community recovery ($\gamma = 0$) is achievable even when the eigenvectors $\boldsymbol{u_2}$ and $\boldsymbol{v_2}$ are not perfectly aligned ($\sin\theta > 0$). This indicates that the spectral method's success depends not merely on eigenvector alignment, but more fundamentally on whether the entry distribution of $\boldsymbol{v_2}$ preserves sufficient structure to enable correct partitioning. In other words, the distributional shape of the eigenvector entries often contains enough information to guarantee perfect classification, even in the presence of some spectral distortion.

# 4 COMPARING THEORETICAL PREDICTIONS WITH SPECTRAL ALGORITHM RESULTS

While the results in Section 3 significantly improve upon the original bounds in Theorem 3.2, all our theoretical analyses rely on the distributional approximation given in Equation 10. As noted previously, this approximation contains errors that, while decreasing as $O(1/\sqrt{n})$, may still affect the accuracy of our predictions for finite sample sizes.

To validate our theoretical framework against the actual spectral algorithm performance, we conduct direct experiments on randomly generated graphs. We generate stochastic block model instances with edge probabilities $a = 0.06n$ and $b = 0.04n$ across a range of graph sizes $n \in \{500, 525, 550, \ldots, 1000\}$. For each instance, we apply our modified **Spectral Partition** algorithm (omitting the degree-based deletion step) and evaluate both the error rate $\gamma$ (comparing the algorithm's partition against the true community structure) and $\theta$ (the angle between the true second eigenvector $\boldsymbol{u_2}$ and the computed approximation to second eigenvector $\boldsymbol{v_2}$).

Furthermore, to provide comprehensive validation across different problem scales, we repeated all the analyses from Section 3 for the complete range of graph sizes $n \in \{500, \ldots, 1000\}$, rather than limiting our evaluation to $n = 500$. These results, including both the Chernoff-based optimization bounds and the Monte Carlo simulation predictions, are consolidated alongside the direct spectral algorithm experiments in Figure 5.

The figure uses opacity to represent graph size, with $n = 500$ shown as nearly transparent points and $n = 1000$ as fully opaque points, creating a visual gradient across problem scales. Different colors distinguish the various analytical approaches:

**Red Points (Theoretical Baseline):** These represent the quadratic bound from Theorem 3.2. Since this bound follows the relationship $\gamma = \sin^2 \theta$, which is independent of $n$, the red points of different opacities overlap completely, forming a single curve.

**Blue Points (Chernoff Analysis):** These show our Chernoff-derived bounds from Section 3.4. As $n$ increases, the achievable frontier moves upward, indicating that the bounds become less tight for larger graphs. This behavior reflects the conservative nature of concentration inequalities for finite sample sizes.

**Green Points (Monte Carlo Simulation):** These represent our normal approximation approach validated through simulation, with 10 repetitions per value of $n$. Similar to the Chernoff bounds, the frontier shifts upward with increasing $n$, particularly in the low-$\gamma$ regime.

**Orange Points and Purple Fit (Direct Algorithm Results):** The orange points show the actual performance of our modified **Spectral Partition** algorithm on randomly generated graphs. To these experimental results, we fit the empirical relationship:

$$\sin \theta = \frac{C}{\sqrt[4]{\log 2 / \gamma}} \tag{13}$$

using OLS regression, with the resulting fitted curve displayed as the purple line.

**Theoretical Significance:** The functional form in Equation 13, combined with the claims of Theorems 2.2 and 3.1, directly yields the final result stated in Theorem 1.3, thus bridging our empirical observations with the theoretical framework.

### 4.1 SCALING BEHAVIOR AND CONVERGENCE ANALYSIS

Several important trends emerge as $n$ increases while maintaining constant ratios $a/n$ and $b/n$. The community detection problem becomes inherently easier for larger graphs, as predicted by both Theorem 1.3 and Theorem 3.2, which allow for smaller error rates $\gamma$ as their left-hand sides increase. This theoretical prediction is confirmed in our results, where larger $n$ values (higher opacity points) consistently achieve lower $\gamma$ values.

More significantly, the gap between the orange points (direct algorithm results) and green points (simulation predictions) of matching opacity decreases with increasing $n$. This convergence validates the error bound which asserts that approximation errors decrease as $O(1/\sqrt{n})$. The observed convergence demonstrates that for large $n$ in the low-$\gamma$ regime, the relationship in Equation 13 and our theoretical prediction in Equation 12 align closely.

This convergence provides strong empirical support for our central claim: **Spectral Partition** alone achieves near information-theoretic performance without requiring the additional **Correction** step, particularly as problem size increases and error rates decrease, precisely the regime most relevant for practical applications.

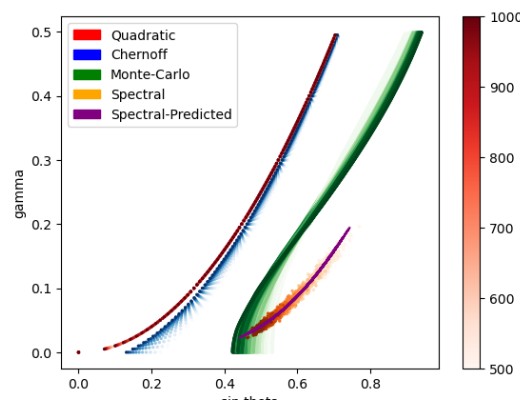

Figure 5: $\gamma$ as a function of $\sin\theta$ for various approaches including experimental results

## 5    CONCLUSION AND FUTURE WORK

We demonstrate that the spectral algorithm achieves near information-theoretic performance, through elimination of degree-based preprocessing and the correction step. Our theoretical analysis through Chernoff bounds, normal approximations, and Monte Carlo validation shows that spectral partition alone can achieve the inverse-logarithmic error rates previously thought to require additional correction steps.

Experimental validation across varying graph sizes confirms that our theoretical predictions become increasingly accurate as the error goes down with $O(1/\sqrt{n})$, with the empirical relationship $\sin\theta = C/\sqrt[4]{\log 2/\gamma}$ bridging our results to established theoretical frameworks. The convergence between multiple analytical approaches in the large-$n$, low-$\gamma$ regime validates our central finding: spectral partition alone suffices for near-optimal community recovery.

These results challenge the assumption that algorithmic complexity improves performance, suggesting instead that careful theoretical analysis can reveal hidden strengths in existing methods. This "less is more" principle may have broader implications for spectral algorithm design.

Several directions emerge from this research: extending our analysis to unbalanced and multi-community cases, analyzing multiple samples derived from the same distributions, developing enhanced inference procedures, investigating computational scaling for massive graphs, analyzing robustness under model misspecification, establishing precise connections to information-theoretic limits, and exploring whether similar simplifications yield improvements in related spectral problems such as graph clustering and matrix completion. The statistical independence between matrix and vector entries preserved by our approach should facilitate these future investigations, as this independence structure can be leveraged for more sophisticated statistical inference and analysis techniques that would be complicated or impossible under the dependencies introduced by traditional preprocessing steps.

## 6    REPRODUCIBILITY STATEMENT

To ensure reproducibility, we provide complete implementation details with specified parameters: graph sizes $n \in \{500, \dots, 1000\}$, edge probabilities $a = 0.06n$ and $b = 0.04n$, and our modified Spectral Partition algorithm that eliminates the degree-based deletion step. Monte Carlo simulations use 50 repetitions for distributional analysis and 10 repetitions for scaling experiments. All random seed numbers are initialized to ensure total reproducibility. Our submitted code includes scripts to regenerate all figures and numerical results, with complete theoretical derivations provided in the appendix.

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

# A   APPENDIX

## A.1   PROOF OF THEOREM 2.2

*Proof.* The matrix $A$ has entries $A_{ij}$ that are sampled from a Bernoulli distribution with success probability $p_{ij}$ where $p_{ij} = a/n$ if $i, j$ belong to the same community, and $p_{ij} = b/n$ otherwise. Therefore, the entries of matrix $M$ have mean zero and variance $\sigma_{ij}^2 = p_{ij}(1 - p_{ij}) \leq \sigma^2$ where $\sigma^2$ is the maximum variance of a single element.

Because $0 < b < a < n/2$ we have:

$$\sigma_{ij}^2 \leq \sigma^2 = \max\left(\frac{a}{n}(1 - \frac{a}{n}), \frac{b}{n}(1 - \frac{b}{n})\right) = \frac{a}{n}\left(1 - \frac{a}{n}\right) \leq \frac{a + b}{n} \tag{14}$$

Let $\lambda_1(M)$ be the largest eigenvalue of $M$. Because $M$ is real-valued and symmetric, $\lambda_1(M) = ||M||$. Now we use the result from Füredi & Komlos (1981) to determine $\mathbb{E}[\lambda_1(M)]$. Since all entries have mean zero and variance at most $\sigma^2$, we have:

$$\mathbb{E}[\lambda_1(M)] = 2\sigma\sqrt{n} + O(n^{1/3}\log n) \tag{15}$$

For large enough $n$, the first term dominates. So $\mathbb{E}[\lambda_1(M)] = O(\sigma\sqrt{n})$. Note: Füredi & Komlos (1981) uses the premise that all entries have mean zero and common variance, but Krivelevich & Vu (2000) showed that the assumption of common variance can be relaxed to $Var[M_{ij}] \leq \sigma^2$.

Next, also according to Krivelevich & Vu (2000), there are positive constants $c$ and $K$ such that for any $t > K$,

$$P\left[|\lambda_1(M) - \mathbb{E}[\lambda_1(M)]| \geq t\right] \leq e^{-ct^2} \tag{16}$$

Combining equations 15 and 16, there is a constant $C_2$ such that for large enough $b$ (and consequently $a, n$), we have with probability $1 - o(1)$:

$$\|M\| \leq C_2 \sigma \sqrt{n} \leq C_2 \frac{\sqrt{a+b}}{\sqrt{n}} \sqrt{n} \tag{17}$$

which completes the proof for Theorem 2.2. □

### A.2 PROOF OF FORMULATION AND PREDICTION FROM SECTION 3.4

#### A.2.1 DERIVING THE MOMENT GENERATING FUNCTION

We start by computing the moment generating function (MGF) for our random variables. Recall that $Y$ represents the difference between two binomial distributions. The MGF of $Y$ is:

$$M_Y(t) = (q_a + p_a e^t)^{n/2}(q_b + p_b e^{-t})^{n/2}$$

Since $-Y$ has MGF $M_{-Y}(t) = M_Y(-t)$, and each entry $X_i$ of our vector is equally likely to be $Y$ or $-Y$, the MGF of $X_i$ becomes:

$$M_{X_i}(t) = \frac{M_Y(t) + M_Y(-t)}{2}$$

$$= \frac{(q_a + p_a e^t)^{n/2}(q_b + p_b e^{-t})^{n/2} + (q_a + p_a e^{-t})^{n/2}(q_b + p_b e^t)^{n/2}}{2}$$

#### A.2.2 APPLYING CHERNOFF BOUNDS

The Chernoff bound gives us:

$$P(X_i \geq a) \leq M_{X_i}(t)e^{-at} \quad \forall t > 0$$

This inequality holds for any positive $t$, but we want to choose the value that gives us the tightest bound. For positive values of $a$, the distribution is dominated by the $Y$ component rather than the $-Y$ component. The optimal choice turns out to be:

$$t^* = \frac{1}{2}\ln\left(\frac{p_a q_b}{q_a p_b}\right)$$

Note that $t^* > 0$ because we assume $p_a > p_b$ (within-community edges are more likely than between-community edges). Substituting this optimal value, we get:

$$P(X_i \geq a) \leq Ce^{-at^*}$$

where the constant $C$ depends only on the model parameters $n$, $a$, and $b$:

$$C = \frac{1}{2}(\sqrt{p_a p_b} + \sqrt{q_a q_b})^{2n} + \frac{1}{2}\left(\sqrt{\frac{q_a^3 p_b^3}{p_a q_b}} + \sqrt{\frac{p_a^3 q_b^3}{q_a p_b}} + q_a q_b + p_a p_b\right)^n$$

#### A.2.3 CONVERTING BOUNDS TO OPTIMIZATION CONSTRAINTS

Now we connect this probabilistic bound to our optimization problem. If $x_i$ is the $i$-th largest element in our sorted vector, and assuming the entries follow the theoretical distribution reasonably well, then

the probability that a random entry exceeds $x_i$ should be approximately $\frac{i}{2n+1}$ (since $i$ entries are larger than $x_i$ out of $2n + 1$ total positions). Therefore:

$$\frac{i}{2n+1} \leq C \cdot e^{-t^* x_i}$$

Solving for $x_i$:

$$x_i \leq \frac{\ln C + \ln(2n + 1) - \ln i}{t^*}$$

For the negative tail (when $i > n$), we use the symmetry of the bounds with $t$ replaced by $-t$, giving us:

$$x_i \geq -\frac{\ln C + \ln(2n + 1) - \ln(2n + 1 - i)}{t^*}$$

### A.2.4 FORMULATING THE COMPLETE OPTIMIZATION PROBLEM

Since all these quantities are known given the model parameters $n$, $a$, and $b$, we can incorporate them into our optimization framework. However, we also need to ensure the resulting vector has unit norm. We introduce the following constraints:

$$x_1^2 + \cdots + x_{2n}^2 \leq 1$$

$$x_{i+1} \leq \frac{\ln C + \ln(2n + 1) - \ln(i + 1)}{\ln C + \ln(2n + 1) - \ln i} x_i \quad \forall i = 1, \ldots, n - 1$$

$$x_i \geq \frac{\ln C + \ln(2n + 1) - \ln(2n + 1 - i)}{\ln C + \ln(2n + 1) - \ln(2n - i)} x_{i+1} \quad \forall i = n + 1, \ldots, 2n - 1$$

### A.2.5 WHY THIS FORMULATION WORKS

Let us elaborate why this setup correctly captures our intentions:

First, regarding the normalization constraint $\sum x_i^2 \leq 1$: We use an inequality rather than equality to make this a convex optimization problem, which can be solved efficiently. However, the optimal solution will automatically satisfy $\sum x_i^2 = 1$. Here's why: if we have a feasible vector $\mathbf{x}$ with $\sum x_i^2 < 1$, we can scale it up by some factor $\lambda > 1$ to get $\lambda \mathbf{x}$ with $\sum (\lambda x_i)^2 = 1$. Since our objective function $\cos \theta$ is positive (by construction) and linear in the entries, scaling up only improves the objective value. Therefore, the optimizer will naturally choose the boundary case where the constraint becomes tight.

Second, regarding the ratio constraints: The Chernoff bounds fundamentally limit how quickly the entries can decay as we move from the largest to the smallest values. The ratio constraints enforce that consecutive entries cannot decay faster than what the Chernoff bounds would allow. Specifically, all entries $x_2, \ldots, x_n$ are constrained relative to $x_1$ through these ratios, and all entries $x_{n+1}, \ldots, x_{2n-1}$ are constrained relative to $x_{2n}$.

If some of these ratio constraints become strict (meaning the actual ratios are smaller than the bounds allow), this doesn't violate our theoretical framework—it simply means the actual distribution has even better concentration than our worst-case analysis predicts. Combined with the normalization argument above, the optimizer will find the largest possible $x_1$ and smallest possible $x_{2n}$ (in absolute value) such that the vector has unit norm, while respecting the decay rates imposed by the Chernoff bounds.

### A.2.6 DERIVING CUMULATIVE SUM APPROXIMATIONS

Starting from our Chernoff-derived bound:

$$x_i \leq \frac{\ln C + \ln(2n + 1) - \ln i}{t^*}$$

We want to approximate the partial sums $s_j = \sum_{i=1}^{j} x_i$. Applying our bound:

$$s_j \leq \sum_{i=1}^{j} \frac{\ln C + \ln(2n+1) - \ln i}{t^*} = \frac{j \ln C}{t^*} + \frac{1}{t^*} \sum_{i=1}^{j} \ln\left(\frac{2 + 1/n}{i/n}\right)$$

For large $n$, we can approximate the discrete sum with a continuous integral:

$$s_j \simeq \frac{j \ln C}{t^*} + \frac{n}{t^*} \int_0^{j/n} \ln\left(\frac{2 + 1/n}{x}\right) dx = \frac{j}{t^*}\left(\ln C + \ln\left(\frac{2n+1}{j}\right) + 1\right)$$

Note that this approximation is only accurate when $j$ is large, which will be the case for our intended application where we consider $j = n - k$ with small $k$.

### A.2.7 APPLYING THE APPROXIMATION TO OUR OBJECTIVE FUNCTION

Recall that our objective function is (from Equation 9):

$$\cos\theta \leq \frac{1}{\sqrt{2n}}\left(\sum_{i=1}^{n-k} x_i - \sum_{i=n-k+1}^{n} x_i + \sum_{i=n+1}^{n+k} x_i - \sum_{i=n+k+1}^{2n} x_i\right)$$

We make two key observations about the optimal solution structure:

First, due to the symmetry of our distribution and constraints, the entries exhibit approximate symmetry around the center: $x_{n+i} \simeq x_{n+1-i}$ for $i = 1, \ldots, n$. This allows us to simplify our expression:

$$\cos\theta \leq \frac{2}{\sqrt{2n}}\left(\sum_{i=1}^{n-k} x_i - \sum_{i=n-k+1}^{n} x_i\right)$$

Second, and more importantly, our Chernoff constraints only establish lower bounds on the ratios $\frac{x_i}{x_{i+1}}$. They don't prevent the optimizer from making some entries arbitrarily small. In particular, nothing stops the optimizer from setting $x_{n-k+1} = \cdots = x_n = 0$ and concentrating all the "budget" (from the normalization constraint $\sum x_i^2 = 1$) into $x_1, \ldots, x_{n-k}$.

Turns out this is exactly what happens in our experiments. The optimizer pushes the middle entries to zero while maximizing the contribution from the largest entries, which doesn't violate our requirement that the entries must be "at least this concentrated" according to the Chernoff bounds.

Therefore, our objective simplifies to:

$$\cos\theta \leq \frac{2}{\sqrt{2n}} \sum_{i=1}^{n-k} x_i = \frac{2}{\sqrt{2n}} s_{n-k}$$

### A.2.8 FINAL THEORETICAL PREDICTION

Since $k = \gamma n$ is small, $n - k$ is large, making our integral approximation valid. Substituting our approximation for $s_{n-k}$:

$$\cos\theta \leq \frac{2}{\sqrt{2n}} s_{n-k} \simeq \frac{2}{\sqrt{2n}} \cdot \frac{(n-k)}{t^*}\left[\ln C + 1 + \ln\left(\frac{2n+1}{n-k}\right)\right]$$

$$= \frac{\sqrt{2n}}{t^*}(1-\gamma)\left(\ln C + 1 + \ln\frac{2 + \frac{1}{n}}{1 - \gamma}\right)$$

Which proves Equation 11. This bound represents our theoretical prediction for the maximum achievable $\cos\theta$ under the Chernoff-derived constraints.

## A.3 PROOF OF FORMULATION AND PREDICTION FROM SECTION 3.5

Under the assumption that the entries $x_i$ follow a standard normal distribution, they effectively partition the probability density function $\phi$ into $2n + 1$ equal quantile intervals. This means:

$$x_i = \Phi^{-1}\left(1 - \frac{i}{2n + 1}\right) = -\Phi^{-1}\left(\frac{i}{2n + 1}\right)$$

The partial sum becomes:

$$s_i = \sum_{j=1}^{i} x_j = -\sum_{j=1}^{i} \Phi^{-1}\left(\frac{j}{2n + 1}\right) = -n\sum_{j=1}^{i} \frac{1}{n}\Phi^{-1}\left(\frac{j/n}{2 + 1/n}\right)$$

For large $n$ and $i$, we can approximate this discrete sum with a continuous integral:

$$s_i \approx -n\int_0^{i/n} \Phi^{-1}\left(\frac{x}{2 + 1/n}\right) dx$$

To evaluate this integral, we use substitution. Let:

$$u = \Phi^{-1}\left(\frac{x}{2 + 1/n}\right)$$

Which means $\Phi(u) = \frac{x}{2 + 1/n}$, so $x = (2 + 1/n)\Phi(u)$ and $dx = (2 + 1/n)\phi(u)du$. When $x = \frac{i}{n}$, we have $u = \Phi^{-1}\left(\frac{i}{2n+1}\right) = -x_i$. Substituting into our integral:

$$s_i \approx -n\int_{-\infty}^{-x_i} u \cdot \left(2 + \frac{1}{n}\right)\phi(u)du = -(2n + 1)\int_{-\infty}^{-x_i} u\phi(u)du$$

Making another substitution $v = -u$ (so $dv = -du$):

$$s_i = (2n + 1)\int_{x_i}^{\infty} v\phi(v)dv$$

Now we can evaluate this integral directly using the substitution $w = v^2/2$ (so $dw = vdv$):

$$\int_{x_i}^{\infty} v\phi(v)dv = \frac{1}{\sqrt{2\pi}}\int_{x_i}^{\infty} ve^{-v^2/2}dv = \frac{1}{\sqrt{2\pi}}\int_{x_i^2/2}^{\infty} e^{-w}dw = \frac{e^{-x_i^2/2}}{\sqrt{2\pi}} = \phi(x_i)$$

Therefore $s_i \approx (2n + 1)\phi(x_i)$. Since both $n$ and $n - k$ are large (with $k$ small), our integral approximation is valid for both $s_n$ and $s_{n-k}$. Accounting for the symmetry in our problem, we have:

$$\cos\theta \leq \frac{2}{\sqrt{2n}}(s_{n-k} - s_n) = \frac{2}{\sqrt{2n}}(2n + 1)(\phi(x_{n-k}) - \phi(x_n))$$

Substituting the explicit expressions:

$$x_{n-k} = -\Phi^{-1}\left(\frac{n - k}{2n + 1}\right) = -\Phi^{-1}\left(\frac{1 - \gamma}{2 + 1/n}\right)$$

$$x_n = -\Phi^{-1}\left(\frac{n}{2n + 1}\right) = -\Phi^{-1}\left(\frac{1}{2 + 1/n}\right)$$

This completes the proof of Equation 12.

