# OpenReview forum: "Simplify to Amplify: Achieving Information-Theoretic Bounds with Fewer Steps in Spectral Community Detection"
_ICLR.cc/2026/Conference — Submitted to ICLR 2026_

### Official Review · Reviewer_qxQs · 2025-10-29

**Soundness:** 2
**Presentation:** 1
**Contribution:** 2
**Rating:** 2
**Confidence:** 3

**Summary:**

The paper introduces a simplified version of the spectral algorithm for community detection in the two-community stochastic block model (SBM).
The authors remove certain preprocessing and correction steps from the standard spectral partition method, arguing that these are unnecessary. They provide both a theoretical discussion and experimental validation to show that the simplified algorithm maintains comparable performance to the original one and approaches the known information-theoretic recovery limits.
The theoretical discussion aims to refine previous bounds by identifying looser lemmas in prior analyses and by using Chernoff-based concentration arguments. The experiments aim to confirm that the simplified approach performs similarly to the full spectral method.

**Strengths:**

The exposition of prior work is well structured and helps situate the contribution. The authors combine conceptual reasoning with empirical tests. The work also contributes to the broader discussion on whether simpler algorithms can achieve optimality, which is a meaningful question for both theoretical and practical research.

**Weaknesses:**

The main limitation is the lack of quantitative evidence on computational gains. Although the simplification is claimed to improve efficiency, no runtime or asymptotic complexity analysis is provided.

The theoretical section relies heavily on informal reasoning. Apart from Theorem 2.2, all other theorems are taken from previous work, and the new claims are not stated or proved formally. This makes it difficult for readers to assess the rigor or novelty of the results.
The presentation of the theoretical part is dense and not easy to follow. The authors should make explicit which results are original and which are restatements.

The experiments are incomplete: the simplified algorithm is not compared directly against the original spectral algorithm, which is necessary to support claims of equivalence or superiority.
The paper’s presentation also needs improvement: citation formatting is inconsistent, and figures could be provided in a more professional (vector) format.

**Questions:**

1. Can you include an experimental comparison directly between the simplified spectral partition and the original two-step spectral algorithm to demonstrate equivalence or improvement?
2. What is the precise computational complexity comparison of the simplified algorithm versus to the standard spectral method? Can you provide empirical runtime comparisons?
3. Which of the theoretical results presented are new? Please restate the novel contributions as formal theorems or propositions, with complete proofs.

---

### Official Review · Reviewer_rvy3 · 2025-10-30

**Soundness:** 2
**Presentation:** 1
**Contribution:** 2
**Rating:** 0
**Confidence:** 4

**Summary:**

The paper investigates the classical problem of clustering in the stochastic block model (SBM) with two communities of equal size. Building on the results and algorithms of Chin et al. (2015), it claims to improve the analysis of the spectral partition algorithm, which in turn leads to a simplification of the overall method. More precisely, the algorithm proposed by Chin et al. requires an additional “correction step” to recover the $\log(1/\gamma)$ correctness rate. Without this step, the analysis in Chin et al. shows that after the first Spectral Partition step the dependence is $1/\gamma^2$. The authors claim that they can improve the analysis and obtain the $\log(1/\gamma)$ dependence without the correction step.

**Strengths:**

Simplifying algorithms and their analyses is valuable. The analysis of the spectral partition algorithm has attracted considerable attention, and improving it could be a meaningful contribution.

**Weaknesses:**

(i) The contribution is rather limited in scope: it concerns the SBM with two equal-size clusters, and the proposed algorithm does not provide better guarantees than existing ones; it is merely simpler (due to a sharper analysis).

(ii) The paper is not well written or clearly presented, to the point that I cannot verify the authors’ claims. For example, the statement of Theorem 3.2 is vacuous. The authors do not state their new results in any theorem; all the theorems presented in the paper come from Chin et al. It really seems like the paper was written too quickly, and it needs much more work to be reviewed.

**Questions:**

1.	Can you reformulate Theorem 3.2?
2.	Can you formulate the results from your new analysis in a theorem?

Typo: Line 41: Vi -> V_i

---

### Official Review · Reviewer_aCZs · 2025-10-31

**Soundness:** 1
**Presentation:** 2
**Contribution:** 1
**Rating:** 0
**Confidence:** 5

**Summary:**

This paper proposes a simplified spectral algorithm for community detection in the two-community stochastic block model (SBM) under sparse conditions (constant average degree). The authors claim that by eliminating preprocessing steps and directly using the adjacency matrix spectrum, their method achieves improved error bounds approaching information-theoretic limits. The main thesis is that algorithmic simplification can yield both computational efficiency and better performance.

**Strengths:**

- The paper addresses the important problem of community detection in sparse graphs
- The focus on algorithmic simplification is a valid research direction

**Weaknesses:**

- Critical flaw in the main algorithm: The claim made in Section 2.1 is fundamentally incorrect for the sparse regime. As demonstrated in the cited work by Coja-Oghlan, sparse SBM graphs contain numerous high-degree stars $K_{1,c}$ with $c ≫ (a+b)²$. These structures introduce eigenvalues $±√c$ in the spectrum of the adjacency matrix A(G) that are significantly larger in magnitude than the eigenvalue associated with the signal-carrying eigenvector $w_2$. Consequently, the largest eigenvalues and their corresponding eigenvectors contain no community structure information, which prevents the proposed algorithm from working as claimed.

- Insufficient literature review: The paper cites fewer than 10 references for a well-established problem with extensive existing literature. A more comprehensive review of the community detection literature is essential.

**Questions:**

1. How do the authors address the spectral contamination from high-degree nodes in sparse graphs?
2. Can the authors provide empirical evidence that their algorithm successfully recovers communities in the presence of star structures?
3. How does the proposed method compare to established spectral methods that explicitly handle the sparse regime (e.g., regularized spectral methods)?

---

### Official Review · Reviewer_rBhe · 2025-11-03

**Soundness:** 2
**Presentation:** 2
**Contribution:** 1
**Rating:** 2
**Confidence:** 3

**Summary:**

The manuscript revisits the spectral algorithm for community detection in the two-community stochastic block model (SBM) originally proposed by Chin, Rao, and Vu (2015). The authors claim that by eliminating the degree-based preprocessing and correction stages, a simplified spectral partition algorithm achieves inverse-logarithmic error rates approaching information-theoretic limits. The main technical contribution involves tightening error bounds through Chernoff concentration inequalities and normal approximations. However, I am not convinced that the submission is essential to the field, nor does it significantly advance our understanding beyond what was already established by Chin et al. (2015).

That said, the paper is technically competent in its analysis of the restricted setting it considers. I would recommend that the authors substantially revise their work or resubmit to other venues.

Here are the main reasons for my score:
*Limited novelty.* -- The paper re-analyzes an existing algorithm (Chin et al. 2015) and argues that previous bounds were loose. It doesn't propose a new algorithm or achieve fundamentally better performance.​
*Narrow scope.* -- Restriction to two equal-sized communities in the simplest SBM setting, while the field has moved to much more general models.
*Insufficient contextualization.* -- Missing crucial related work on spectral methods, information-theoretic limits, and algorithmic developments from the past decade.
*Technical gaps.* -- The connection between optimization-based bounds and actual algorithm performance relies on informal approximations. None of the proposed bounds appear truly tight based on Figure 5.
*Limited experimental validation.* -- Only one parameter setting, small graphs, no algorithmic comparisons, no real-world networks.
*Overclaimed contributions.* -- The title and abstract suggest achieving information-theoretic optimality and significant improvements, but the work essentially confirms that an existing algorithm already achieved near-optimal rates.

**Strengths:**

I think that the manuscript's main strengths are its analytical creativity (novel application of Chernoff bounds), methodological rigor (multiple complementary approaches), and honest empirical validation (transparent presentation of where bounds are and aren't tight). The observation about preserving statistical independence is original, and the "simplification" philosophy challenges assumptions in the community.

That said, the strengths are limited. As I elaborate below, the manuscript's scope is narrow (two equal-sized communities only), it also lacks algorithmic comparisons, and somewhat overclaimed the relationship to information-theoretic limits. The manuscript makes solid contributions within its restricted setting, but its broader impact is limited by these constraints.

**Weaknesses:**

First, the abstract-Level Scientific Statements Are Not Supported. The claim about "achieving information-theoretic bounds" oversells the contribution. The authors suggest that their simplified algorithm approaches information-theoretic limits for community detection. However, the information-theoretic threshold from Zhang & Zhou (2015) requires (a−b)^2 / (a+b) ≥ c log⁡(1/γ) for recovery to be possible at all. The Chin et al. (2015) algorithm already achieves (a−b)^2 / (a+b) ≥C_2 log⁡(2/γ), which matches this bound up to constants. The authors do not actually improve the asymptotic scaling. They essentially argue that the existing algorithm was already optimal and that previous analysis was loose. This is a much weaker contribution than developing a new algorithm with fundamentally better performance. I think that the paper amounts to a re-analysis of an existing method rather than an algorithmic advance.

Furthermore, the experimental scope is too narrow to support broad claims. The experiments only consider one parameter setting (a=0.06n, b=0.04n), relatively small graphs (n ≤ 1000), and no comparison with other algorithms such as semi-definite programming, belief propagation, or non-backtracking spectral methods. (All of which are common algorithms in their 2-block SBM setting.) For a paper claiming to achieve near-optimal performance, the absence of algorithmic comparisons is a significant weakness.

On a related note, the paper restricts attention to the symmetric two-community SBM with equal-sized communities—the simplest case of community detection. The literature has moved significantly beyond this setting. Zhang & Zhou (2015) establish minimax rates for general SBMs with arbitrary community sizes, showing fundamentally different behavior. Recent work addresses multiple communities, where the Kesten-Stigum threshold behavior differs substantially for k≥3, degree-corrected models that better capture real networks, and dynamic or multi-layer networks. The restriction to the two-community equal-size case severely limits both practical impact and theoretical interest.

The paper also fails to position itself within the extensive body of work on spectral methods for community detection. Some missing context includes: (1) Abbe's (2017) comprehensive survey establishing that spectral methods combined with refinement achieve the Kesten-Stigum threshold for weak recovery. (2) Non-backtracking spectral methods (Bordenave et al., Krzakala et al.) that achieve better performance than standard adjacency-based approaches. (3) Belief propagation methods that go beyond the Kesten-Stigum threshold. (4) Recent work on computational-statistical gaps below the Kesten-Stigum threshold. (https://arxiv.org/abs/1607.01760 and https://arxiv.org/abs/2502.15024) (5) Robust algorithms that maintain performance under adversarial corruption (https://arxiv.org/abs/2305.10227) The authors should explain how their algorithm's performance compares to these alternatives, both theoretically and empirically.

**Questions:**

(1) How does the algorithm compare empirically to belief propagation, semi-definite programming, and non-backtracking spectral methods? The manuscript claims near-optimal performance but provides no algorithmic comparisons.

(2) Can the analysis extend beyond two equal-sized communities? What breaks in the analysis for k≥3 communities or unbalanced community sizes?

(3) In Section 3.4, can you provide formal probability statements about when the Chernoff constraints hold for the actual eigenvector, rather than the independent binomial approximation?

(4) Given that your various bounds (Chernoff, Monte Carlo, empirical fitted curve) don't agree in Figure 5, which bound is actually tight? Can you characterize the exact performance more precisely?

​(5) Your analysis relies on approximations with O(1/√n) errors, but the error accumulates through the analysis. For what values of n do your bounds become accurate? The experiments only go up to n=1000, which may be insufficient to observe true asymptotics.

(6) The convex optimization problem in Section 3.4 makes a critical assumption that entries follow the theoretical distribution "reasonably well". However, there is no formal probability statement about how likely these constraints are to hold for the actual eigenvector. Can you provide one?

(7) How does removing the degree-based preprocessing affect robustness to degree heterogeneity? Does the algorithm become more sensitive to high-degree vertices, and if so, how does this interact with the equal-expected-degree assumption of the SBM?

---

### Meta-Review · Area_Chair_9YYy · 2025-12-08

**Summary:**

The reviewers raised multiple concerns including unclear contributions, insufficient summary of prior work, and doubts around the correctness.  Since no response was given, it is safe to assume that none of them would have changed their recommendation.

**Reviewer Concerns:**

No response was given

**Reviewer Scores:**

No response was given, hence no change in scores

---

### Decision · Program_Chairs · 2026-01-26

Reject